# Resuscitation and Initial Management After Moderate-to-Severe Traumatic Brain Injury: Questions for the On-Call Shift

**DOI:** 10.3390/jcm13237325

**Published:** 2024-12-02

**Authors:** Jesús Abelardo Barea-Mendoza, Mario Chico-Fernández, Maria Angeles Ballesteros, Alejandro Caballo Manuel, Ana M. Castaño-Leon, J. J. Egea-Guerrero, Alfonso Lagares, Guillermo Morales-Varas, Jon Pérez-Bárcena, Luis Serviá Goixart, Juan Antonio Llompart-Pou

**Affiliations:** 1Trauma and Emergency ICU, Critical Care Deparment, 28041 Madrid, Spain; 2Instituto de Investigación Sanitaria Hospital 12 de Octubre (imas12), 28041 Madrid, Spain; 3Trauma and Neurocritical ICU, Service of Intensive Care, Hospital Universitario Marqués de Valdecilla-IDIVAL, 39008 Santander, Spain; mdelosangeles.ballesteros@scsalud.es; 4School of Medicine, Universidad de Cantabria, 39011 Santander, Spain; 5Department of Neurosurgery, Hospital Universitario 12 de Octubre, 28041 Madrid, Spain; 6Neurocritical Care Unit, Hospital Universitario Virgen del Rocío, 41013 Sevilla, Spain; juanjo.egea.sspa@juntadeandalucia.es; 7Departamento Ciencias de la Salud, Universidad Loyola Andalucía, 41704 Sevilla, Spain; 8Departamento de Cirugía, Facultad de Medicina, Universidad Complutense de Madrid, 28040 Madrid, Spain; 9Trauma and Neurocritical Care ICU, Hospital Toledo, 45005 Toledo, Spain; gmoralesv@sescam.jccm.es; 10Trauma and Neurocritical Care ICU, Hospital Universitari Son Espases, 07010 Palma, Spain; juan.perez@ssib.es (J.P.-B.); juanantonio.llompart@ssib.es (J.A.L.-P.); 11Institut d’Investigació Sanitària Illes Balears (IdISBa), 07010 Palma, Spain; 12Intensive Care Department, Hospital Universitari Arnau de Vilanova, Av. Alcalde Rovira Roure, 80, 25198 Lleida, Spain; lservia.lleida.ics@gencat.cat; 13IRBLLeida (Lleida Biomedical Research Institute’s Dr. Pifarré Foundation), Av. Alcalde Rovira Roure, 80, 25198 Lleida, Spain

**Keywords:** traumatic brain injury, resuscitation, secondary brain injury, multimodal monitoring, coagulopathy, hemorrhagic shock

## Abstract

Traumatic brain injury (TBI) is a leading cause of disability and mortality globally, stemming from both primary mechanical injuries and subsequent secondary responses. Effective early management of moderate-to-severe TBI is essential to prevent secondary damage and improve patient outcomes. This review provides a comprehensive guide for the resuscitation and stabilization of TBI patients, combining clinical experience with current evidence-based guidelines. Key areas addressed in this study include the identification and classification of severe TBI, intubation strategies, and optimized resuscitation targets to maintain cerebral perfusion. The management of coagulopathy and special considerations for patients with concomitant hemorrhagic shock are discussed in depth, along with recommendations for neurosurgical interventions. This article further explores the role of multimodal neuromonitoring and targeted temperature management to mitigate secondary brain injury. Finally, it discusses end-of-life care in cases of devastating brain injury (DBI). This practical review integrates foundational and recent advances in TBI management to aid in reducing secondary injuries and enhancing long-term recovery, presenting a multidisciplinary approach to support acute care decisions in TBI patients.

## 1. Introduction

Traumatic brain injury (TBI) constitutes one of the leading causes of disability and mortality worldwide. It results from external mechanical forces applied to the head, leading to primary injury at the moment of impact and often followed by secondary injury as a consequence of physiological responses [1]. TBI is a medical emergency, requiring prompt management to prevent secondary brain damage and improve patient outcomes. The complexity of TBI relies on the diversity of clinical presentations and the unpredictable course of injury progression. In the acute setting, the initial management of moderate-to-severe TBI is crucial to stabilize the patient, reduce secondary injury, and maintain homeostasis. This often involves advanced airway management, resuscitation to maintain hemodynamic stability, and the use of multimodal monitoring to guide therapeutic decisions. Despite advancements in monitoring and treatment, the variability of injuries and individual patient factors make standardized care difficult, and guidelines are often based on limited evidence. Given the high risk of deterioration in these patients, early identification of predictors for poor outcomes and the implementation of targeted therapies are essential for improving prognosis and reducing long-term disability. This text aims to combine the best available evidence with bedside clinical experience in response to common challenges in the early hours following moderate-to-severe TBI.

## 2. Questions for the On-Call Shift

### 2.1. Identification of Severe or Complicated TBI: Scores and Predictors of Deterioration

TBI occurs as a result of the application of external forces [1]. These forces cause a primary injury, followed by a secondary injury. In the intensive care unit (ICU) setting, the most widely used classification system for TBI is the Glasgow Coma Scale (GCS), which categorizes patients into mild, moderate, and severe. The GCS should be reported based on its three individual components, with special emphasis on the motor response, which holds the greatest prognostic value. The patient’s age and pupillary status, along with ethanol and drug levels, should be included to complete this assessment. This approach can under- or over-estimate prognosis [2]. This is likely due to the heterogeneity of injuries, clinical presentations, and common factors that can influence outcomes, such as agitation, seizures, medications, and substance abuse. In an attempt to improve accuracy, radiological scales have been used to characterize the type of injury [3]. In this context, a stratification system that incorporates clinical evaluation, imaging (Marshall, Rotterdam, and Helsinki classifications), and ideally, biomolecular markers (mainly glial fibrillary acidic protein (GFAP) and ubiquitin C-terminal hydrolase L1 (UCH-L1)) could help to further individualize treatment. A group of particular interest is formed by patients with moderate TBI. They should be considered at high risk of neuroworsening, requiring imaging studies and close neuromonitoring [4,5]. Different factors have been associated with a worse prognosis in moderate TBI, such as age, shock, hemorrhage, coagulopathy, the type of brain injury, and thoracic trauma [6]. A risk-based stratification system integrating key parameters and clinical monitoring for neuroworsening rather than fixed categories can be useful (Table 1) [2,5].

### 2.2. Intubation in Brain Trauma: Who, How, and When?

Intubation following TBI is a high-risk procedure, with the absence of conclusive guidelines to guide its management. Every intubation in a trauma patient should be considered a difficult airway. TBI patients often present physiological and anatomical challenges during intubation. The intubation process can lead to dramatic increases in secondary damage due to hypoxia, hypotension, and elevated intracranial pressure (ICP). The main indications for intubation are loss of bulbar reflexes, hypoxemia, or hypoventilation, and a low level of consciousness. Although a GCS of <9 has been commonly used as a threshold, a fixed limit is not recommended [7,8]. Professionals who perform intubation should be equipped to thoroughly assess the patient’s injuries while being proficient in advanced airway management following standardized practices. Experience and training are key to safely managing the emergency airway, since alternative methods, such as a laryngeal mask or surgical airway, should be employed to ensure oxygenation if necessary [9]. Achieving hemodynamic and ventilatory stability throughout the intubation process is critical. Depending on the patient’s condition, C-ABC resequencing (Catastrophic compressible hemorrhage—Airway, Breathing, and Circulation) may be necessary to prevent hemodynamic collapse. This involves the early use of vasopressors, availability of blood products, and rapid screening for associated injuries (i.e., pneumothorax), prioritizing circulation over intubation [10]. A shock index (SI) greater than 0.8 has been suggested as the threshold for this approach [8]. The use of sedatives that provide greater stability should be prioritized (midazolam or ketamine). Adding opioids can reduce ICP during airway manipulation. Most patients will require immobilization due to suspected cervical injury, complicating intubation. In these patients, the use of a bougie or stylets alongside direct laryngoscopy or videolaryngoscopy may increase the success rates [11]. A ramped position may reduce the rise in ICP and the risk of bronchoaspiration [12,13]. The use of capnography during bag ventilation (to assess airway patency) and later to confirm adequate intubation is highly recommended [14]. The HEAD bundle for brain trauma intubation is provided in Figure 1.

### 2.3. What Are the Resuscitative Treatment Goals for This Patient?

Blood pressure, cerebral perfusion pressure (CPP), and cerebral blood flow are interdependent variables that shape cerebral perfusion. Minimizing hypovolemia and hypotension are essential in preventing secondary brain damage [15]. Current Brain Trauma Foundation (BTF) guidelines recommend different systolic blood pressure thresholds (100 and 110 mmHg depending on clinical characteristics, including age) during initial resuscitation [16]. However, recent findings showing lower mortality rates in TBI patients with systolic blood pressure > 110 mm Hg regardless of age and gender may challenge this recommendation [17,18]. Given recent reports indicating higher mortality with balanced solutions in patients with acute brain injury, saline 0.9% should remain the resuscitation fluid of choice until new evidence emerges [19]. Current BTF guidelines recommend CPP ranges between 60 and 70 mmHg, avoiding values below 50 mmHg [16]. However, it must be kept in mind that the relationship between CPP and cerebral blood flow will not be linear when autoregulation is disturbed. Cerebrovascular autoregulation (CA) plays a major role in the pathophysiology of adult severe TBI. There is no consensus on how to incorporate it in clinical decisions, but PRx (Pressure Reactivity Index) is the most accepted method [20]. The Phase II COGiTATE trial demonstrated the feasibility and safety of individualized CPP management guided by cerebral autoregulation [21]. The authors utilized the PRx, derived from the ICP and arterial pressure monitoring, to adjust CPP based on the concept of optimal CPP (defined as the CPP value corresponding to the lowest PRx value). While robust evidence is still lacking, this approach appears promising in tailoring optimal CPP ranges and potentially improving outcomes [22].

Cerebral blood flow is important to meet the metabolic demands of the brain. Under normal conditions, PaCO_2_ is the main determinant of cerebral blood flow. Guidelines recommend PaCO_2_ between 35 and 45 mm Hg due to the risk of inducing ischemia. Moderate hyperventilation can be performed as an extraordinary measure in patients at risk of herniation. Both hypoxemia and hyperoxia have been shown to increase secondary damage [23]. Recent guidelines suggest maintaining pO_2_ levels between 80 and 120 mmHg [24]. TBI itself causes an intrinsic increase in metabolism. Feeding patients to achieve basal caloric replacement at least on the fifth day after TBI is recommended to reduce mortality. There is insufficient evidence on the influence of vitamins and supplements [16]. It is recommended to monitor osmolarity and sodium levels. Typical objectives include serum sodium of 145 to 155 mmol/L and/or serum osmolality of 310 to 320 mOsm/kg, although the correlation between these values and ICP is questionable [25].

### 2.4. How Should We Manage Patients with Concomitant Hemorrhagic Shock and Traumatic Brain Injury?

Hemorrhagic shock and TBI are the leading causes of morbidity and mortality in severe trauma. The coexistence of both in the same patient presents unique clinical challenges. In a study from the Spanish Intensive Care Trauma Registry (RETRAUCI), both conditions were observed in 3.8% of the patients. Notably, these patients had a mortality rate of 40%, with intracranial hypertension being the leading cause [26]. In a recent study including 564 patients with TBI and hemorrhagic shock evaluating the effect of whole blood transfusion, overall 30-day mortality was 42% (237/564). Death was attributed to TBI in 100/237 cases (42.2%) [27].

If the patient presents significant hemodynamic instability, this will be the first issue to be addressed through resuscitation and bleeding control [28]. In the meantime, a strategy of careful, brain-focused resuscitation should be followed. This should include using higher blood pressure targets (SBP > 110 mmHg), avoiding hypotension, judicious use of sedation, and maintaining oxygenation, alongside aggressive correction of coagulopathy. If the patient needs to be transferred for hemorrhagic source control, neurocritical care should not be neglected, and hyperosmolar agents or mild hyperventilation might be used. Transcranial Doppler (TCD) and optic nerve sheath diameter (ONSD) non-invasive monitoring are justified during the initial resuscitation to reasonably rule out significant intracranial hypertension and select patients that would benefit from invasive monitoring during procedures (radiology lab or theater) when available and feasible. This would help to minimize secondary brain injury and individualize therapeutic targets. Although extremely rare (<2% according to series), some patients may present with injuries requiring emergent interventions that compete for priority (e.g., intracranial lesion and hemoperitoneum) [29]. In these cases, the presence of hybrid rooms with access to endovascular and open techniques, including on-site imaging, can improve the workflow [30]. In this context, clinical leadership is essential to establish priority in treating these injuries. There are no universal recommendations; therefore, it is essential to integrate the patient’s physiology (how unstable they are), diagnosis (causes of the instability), and neurological status (GCS, pupil condition, imaging). Simultaneous surgery on multiple cavities is an option [31]. This requires modifying the patient’s positioning in the operating room and proper team distribution.

### 2.5. Coagulopathy in Traumatic Brain Injury: Diagnosis and Treatment

Coagulopathy is a critical complication in severe TBI, associated with increased mortality and worse neurological outcomes. It is characterized by the massive release of tissue thromboplastin and systemic activation of coagulation, along with excessive fibrinolysis [32]. Recent studies indicate that coagulopathy in TBI correlates with injury severity as well as factors like cerebral hypoperfusion, hypoxia, and systemic inflammation, which exacerbate neuronal damage and increase the risk of secondary intracranial hemorrhage [33]. We must adhere to general guidelines, avoiding hypothermia and acidosis [34]. Early identification of coagulopathy through traditional and viscoelastic tests is crucial for optimizing therapeutic strategies [35]. During the initial resuscitation phase, we recommend that coagulation parameters should be reviewed every 4–6 h. Subsequently, if the patient has stabilized, monitoring should be performed every 12–24 h. In patients with TBI who develop coagulopathy, the use of fresh frozen plasma (FFP) or prothrombin complex concentrate (PCC) is recommended depending on the clinical situation. FFP (15 to 20 mL/kg) is typically used to correct coagulopathy when the INR exceeds 1.5 or in cases of significant bleeding. However, its large volume and the time needed for preparation can be limiting [36]. The effect of FFP may be transient, especially if the underlying cause of coagulopathy persists or if there is ongoing loss of coagulation factors due to active bleeding. Therefore, repeated doses of FFP are often required to maintain adequate hemostasis. Alternatively, PCC (25 to 50 IU/kg), being more concentrated and faster to administer, is particularly effective for patients requiring immediate correction, especially those on prior anticoagulation therapy. Recent studies suggest that PCC may improve survival and reduce mortality compared to FFP, though its use should be guided by viscoelastic tests like ROTEM or TEG to tailor the treatment and maximize clinical benefits while minimizing thrombotic risks [37]. The administration of platelets in TBI aims to correct platelet dysfunction and is recommended for patients with significant intracranial hemorrhages, involving the transfusion of one to two units of platelets. However, evidence on its effectiveness in reducing mortality is mixed. Therefore, platelet transfusion should be evaluated on a case-by-case basis, incorporating specific platelet function tests. This approach is particularly important in patients who are scheduled to undergo neurosurgical procedures [38]. Tranexamic acid (TXA) should be administered to patients with mild-to-moderate traumatic brain injury (TBI) within the first three hours after trauma. Its early use has been shown to reduce mortality by decreasing the risk of intracranial hemorrhage progression. The recommended dose is 1 g intravenously in a bolus, followed by 1 g in continuous infusion over 8 h. The efficacy of TXA decreases after three hours, making early administration crucial. According to the landmark CRASH-3 randomized controlled trial, the risk of head injury-related death was reduced with tranexamic acid in patients with mild-to-moderate head injury (RR 0.78 [95% CI 0.64–0.95]) but not in patients with severe head injury (0.99 [95% CI 0.91–1.07]; *p* value for heterogeneity 0.030), so its use is not routinely recommended in cases with severe TBI [39,40]. Understanding the underlying molecular mechanisms and signaling pathways involved in coagulation dysfunction in severe TBI remains an important goal for developing more effective and personalized therapies [41]. In this complex context, there are also controversies and questions remaining regarding the timing, dosing, and agent to use for venous thromboprophylaxis in TBI, a relevant issue in immobilized patients that may have received TXA [42]. The 2016 version of the BTF guidelines states that there is insufficient evidence to support a Level I or II recommendation for treatment of deep venous thrombosis (DVT) in severe TBI patients. Low molecular weight heparin (LMWH) or low-dose unfractioned heparin may be used in combination with mechanical prophylaxis. However, there is an increased risk for expansion of intracranial hemorrhage. In addition to compression stockings, pharmacological prophylaxis may be considered if the brain injury is stable, and the benefit is considered to outweigh the risk of increased intracranial hemorrhage. There is insufficient evidence to support recommendations regarding the preferred agent, dose, or timing of pharmacologic prophylaxis for DVT [16].

### 2.6. How Should We Manage TBI Patients Who Are on Antiplatelet and Anticoagulant Therapy?

The use of antithrombotic therapy is recognized as a negative prognostic factor in the clinical course of patients with TBI. Specific phenotypes have been identified. Age and anticoagulant use correlate with poorer outcomes, even among patients exhibiting high scores on the GCS [43]. Guidelines recommend the prompt reversal of anticoagulant therapy, while recommendations regarding the management of antiplatelet therapy remain less conclusive [34,44]. Reversal of vitamin K antagonists can be guided by the International Normalized Ratio (INR), whereas the management of newer anticoagulants depends on the time elapsed since the last dose of the medication. Reversal should be undertaken using specific antidotes; in the absence of these, PCC is advised (Figure 2).

The impact of prior use of antiplatelet agents (APAs) on the outcomes of TBI patients remains controversial. Although a higher incidence of intracranial hemorrhagic lesions following trauma has been documented, it has not been established that prior use of APAs influences the need for surgical evacuation, length of hospital stay, or mortality [45]. Currently, the use of agents such as desmopressin and platelet transfusions is not recommended routinely [34]. Therefore, the indication for platelet transfusion in patients with TBI is reserved for those with thrombocytopenia (less than 100 × 10^9^/L) and for patients requiring urgent neurosurgical procedures [34,44,46]. The increasing utilization of novel APAs, along with their administration in combination with other antiplatelet and/or anticoagulant agents, raises new questions that warrant further investigation. In this context, it has been reported that in patients receiving APAs and exhibiting platelet dysfunction assessed through thromboelastography or aggregometry, platelet transfusion may enhance survival in those with severe TBI [47].

### 2.7. Imaging: What Tests and When Are They Needed?

Non-contrast head CT is the most widely used neuroimaging technique in TBI patients. The American College of Radiology recommends acquiring a non-contrast head CT for all patients who suffer an acute moderate-to-severe TBI to evaluate structural damage [48]. Non-contrast CT is sensitive and specific for the presence of intracranial hemorrhage, extra-axial fluid collections, skull fractures, and radiopaque foreign bodies [48]. Non-contrast CT also detects cerebral contusions, cerebral edema, swelling, and signs of herniation. In suspected DAI (unexplained neurological findings despite “negative” non-contrast CT), the potential corpus callosum or brainstem involvement makes magnetic resonance imaging (MRI) the elective imaging technique to diagnose and characterize the extent of DAI. After initial scanning, it is also recommended to perform a follow-up CT in any TBI patient presenting clinical neuroworsening, despite an initial early CT showing a minimal (or even absent) injury. New lesions appear in up to 16% of patients with diffuse lesions and in 25–45% of the patients presenting with cerebral contusions. Most injuries progress within 6–9 h after the initial injury, occurring more frequently if the initial non-contrast CT is performed within 2–3 h after TBI. A recent meta-analysis showed that subarachnoid hemorrhage, subdural hemorrhage, epidural hemorrhage, and contrast extravasation had a predictive effect on the occurrence of hemorrhagic progression of the contusions [49]. The presence of a traumatic vascular lesion has been detected in up to 17% of patients with cervical traumatic injuries and in those with skull base injuries affecting the carotid canal. The American College of Radiology recommends performing CT–angiography in patients with TBI to rule out the presence of a traumatic vascular lesion when the patient presents clinical risk factors or positive findings, as listed by the modified Denver criteria [50]. A practical approach includes its performance in the immediate follow-up CT. The performance of CT–angiography is similar to that obtained with four-vessel angiography, traditionally considered the reference standard, which is especially useful when CT–angiography is inconclusive or when endovascular intervention is considered [50]. Additionally, traumatic venous injury can occur, although it is commonly overlooked. From the imaging point of view, the most important risk factor for traumatic venous injury is a skull fracture (or less commonly, a penetrating foreign body) that involves a dural venous sinus or jugular bulb/foramen [48]. In these cases, the performance of a venous CT is recommended [48].

In cases of TBI with concurrent severe hemodynamic instability in the context of life-threatening hemorrhage, we must prioritize the precise and early control of hemorrhage with associated restoration of circulating blood volume. Uncontrolled cases may require simultaneous imaging and multisystem surgery in hybrid rooms, being the main objective the control of bleeding and the avoidance/minimization of secondary brain insults [51].

Initial neuroimaging workflow in moderate-to-severe TBI can be found in Figure 3.

### 2.8. Emergent Neurosurgery: Indications and Controversies

Intracranial hematomas are present in 25% of patients with TBI who require hospital admission and up to 45% of patients with TBI who require ICU admission [52]. Guidelines for the surgical management of TBI patients have not been updated since their publication, and the recommendations were based on case series [53]. However, the natural history of most intracranial hematomas or high ICP states precludes the design of randomized controlled trials in many circumstances. In general, surgical evacuation is recommended if the hematoma volume is >25–30 cc for epidural hematomas and brain contusions or thickness > 10 mm for acute subdural hematomas (SDH). Although these recommendations are made regardless of the patient’s level of consciousness, as clinical deterioration is expected without intervention, patients with brain atrophy may tolerate such volumes. Therefore, additional features, such as signs of mass effect and the patient’s neurological condition, must be considered. Lesions without significant mass effect and in patients with stable clinical conditions can often be managed conservatively. On the other hand, for hematomas with smaller volumes, surgical evacuation is still recommended if there is a midline shift >5 mm or effacement of the basal cisterns and/or in case of neurological worsening not explained by extracranial reasons. This scenario is commonly associated with the location of EDH or ICH (e.g., temporal region) or with SDH, which can contribute to ischemic changes in the affected hemisphere due to local mass effects.

Surgery for hematomas located in the posterior fossa is indicated if there is distortion, or obliteration of the fourth ventricle, compression of the basal cisterns, or the presence of obstructive hydrocephalus. Patients with TBI may develop brain swelling and refractory high ICP, even when the hematoma is evacuated. Secondary decompressive craniectomy (DC) can be helpful to decrease mortality when used after exhausted other treatment options. However, the benefit in mortality can be translated into additional survivors with disability [54]. The pre-emptive decision to not replace the bone flap after SDH evacuation (primary DC) seemed not to be associated with a reduction in mortality or a better functional outcome [55]. The placement of an external ventricular catheter (EVD) is useful for reducing ICP. However, the risk of EVD-related infection, difficulties related to small ventricles, and coagulopathy must be addressed before indication [56]. Surgical decision-making is a complex process in which additional characteristics should be accounted for, including the patient’s age, comorbidities, antithrombotic treatment, frailty, pupillary asymmetry, timing from clinical deterioration, and presence of severe hypoxia or shock, which are important factors influencing the patient’s outcome. Additionally, reviewing the patient’s living will, either through official documentation or consultation with relatives, is essential, especially before recommending a surgical treatment that could result in survival with significant disability.

### 2.9. Multimodal Neuromonitoring: Who and How?

Multimodal neuromonitoring is one of the most frequent interventions in patients with severe TBI. It enables targeted treatment to minimize secondary injury and manage intracranial hypertension or cerebral hypoxia. Key questions to ask are as follows: which patients benefit; how do we guide treatment; and when should monitoring be discontinued? Indications should consider the patient’s GCS, head CT findings, and systemic condition. There is agreement on monitoring ICP in patients with a GCS < 9 after resuscitation and pathological CT (16). In those with GCS < 9 and low-risk CT (subarachnoid hemorrhage, intraventricular hemorrhage, petechiae), close follow-up with repeat CT and use of non-invasive monitoring (ONSD or TCD) may be considered. This approach is particularly recommended when CNS depressors (i.e., alcohol) are suspected of playing a role in the patient’s neurological responses. Some of the most difficult patients are those with GCS 9–13 and injuries that do not meet criteria for surgical evacuation. These patients are usually difficult to assess clinically (neurological fluctuation and restlessness) and do not have a clear indication for invasive monitoring. However, they may benefit from such invasive monitoring if risk factors for clinical deterioration or systemic conditions are present, for example, in those patients who do not clearly maintain their airway patency, requiring sedation and mechanical ventilation [57]. Regarding coagulation, INR < 1.6 combined with viscoelastic testing could be safe and can reduce the time to monitoring [58]. Non-invasive neuromonitoring techniques can assist in decision-making [59]. The use of PbtO_2_ has been shown to reduce episodes of cerebral hypoxia, which may occur in patients with normal ICP [60]. Indications include decompressive craniotomy, high ICP, vascular injury, or extracranial injuries (i.e., aortic injury and hypoxemia) that compromise cerebral perfusion and oxygenation. These indications may broaden as ongoing trials progress (ACTRN12619001328167P; NCT03754114); however, it is important to note that the use of PbtO_2_ in combination with ICP monitoring has not demonstrated improvement in patient outcomes in recent clinical trials to date [61]. The Seattle International Severe Traumatic Brain Injury Consensus Conference (SIBICC) guidelines define three profiles requiring different interventions: patients with isolated intracranial hypertension, isolated cerebral hypoxia, or both. Interventions should be selected based on their effectiveness and balanced against side effects [56]. Therapeutic goals should be flexible and adapted to the patient’s physiology (e.g., ICP of 25 mmHg and normal PbtO_2_). The potential benefit of multimodal monitoring may stem from the strategies we adopt based on data interpretation rather than solely from the placement of the monitor. Nevertheless, it is essential to highlight the debate about the effect of neuromonitoring on patient outcomes. This controversy leads to significant variability and the adoption of clinical protocols for managing patients without invasive monitoring [62]. Tools to objectively assess treatment intensity (the Therapy Intensity Level Scale) are valuable for tracking progress and decision-making [63]. Daily review is essential to determine when to discontinue monitoring. Discontinuation is recommended after at least 48 h of meeting treatment targets and stable CT findings. It is recommended to wait at least 12 h between prophylactic LMWH administration and the probe withdrawal.

### 2.10. Analgesia and Sedation: Best Decision in Acute Care of Traumatic Brain Injury

Analgesia and sedation are essential neurocritical care interventions, both at basic and advanced levels (e.g., barbiturates). Proper assessment of TBI patients through tools like the bispectral index (BIS), nociception level index (NOL^®^), and neuromuscular monitoring with the train of four (T.O.F.) is crucial for quality care [16,64]. The choice of the initial sedation and analgesia should be based on the pharmacokinetic and pharmacodynamic properties of the drugs used (see Table 2).

Propofol and midazolam are first-line drugs, with the choice depending on the patient’s clinical condition. Midazolam should be reserved for cases of significant hemodynamic instability [66]. Dexmedetomidine and ketamine could be used as adjuvants, but they are not recommended as monotherapy. Recent studies have shown that dexmedetomidine may have a neuroprotective effect. Xu et al. showed improved survival in TBI patients with dexmedetomidine use [67]. While ketamine was traditionally contraindicated in patients with intracranial hypertension, recent research has shown its safety and even some neuroprotective effects, but its efficacy should be confirmed in the ongoing trials [68] (NCT05097261). Barbiturates are now reserved for refractory intracranial hypertension and they are not recommended as first-line sedatives. Among these, thiopental has demonstrated superior efficacy in treating intracranial hypertension [69]. Although inhaled sedation may offer certain advantages, it should not be used in TBI patients due to its vasodilatory effects with variable influence on the ICP [70]. Adequate analgesia, using both opioid and non-opioid drugs, is crucial to reduce nociceptive stimuli that may increase ICP. Fentanyl is recommended when ICP control is needed, while remifentanil, a short-acting opioid, is preferred for patients in whom sedation withdrawal is planned. Additionally, antipsychotic medications (such as haloperidol or quetiapine) should be used to manage anxiety or delirium [64]. The reduction and eventual withdrawal of sedation is a critical moment in managing TBI patients. Early discontinuation may trigger intracranial hypertension, leading to the harmful effects of secondary injury, while delayed withdrawal may increase the risk of complications such as infections, ICU-acquired weakness, delirium, and prolonged ICU stay [71]. The SIBICC guidelines recommend resuming safely by considering the initial severity of TBI, the stability of ICP values, and the absence of the need for interventions for at least 24 h [72].

Seizure prophylaxis constitutes another matter of debate in moderate-to-severe TBI. Recent evidence and recommendations from the Neurocritical Care Society following a population, intervention, comparator, and outcome (PICO) approach suggest that, based on GRADE criteria, prophylactic antiseizure medications (ASM) or no ASM may be used in patients hospitalized with moderate-to-severe TBI (weak recommendation and low quality of evidence). When ASM are used, they suggest using levetiracetam over phenytoin/fosphenytoin (weak recommendation, very low quality of evidence) for a short duration (≤7 days, weak recommendation, and low quality of evidence) [73]. These recommendations are different from those from the 2016 Brain Trauma Foundation guidelines but are aligned with current clinical practice in the U.S. [16,74].

### 2.11. Not Too Cold, Not Too Hot: Targeted Temperature Management

Hypo- and hyperthermia have been consistently associated with worse clinical outcomes in patients with TBI [75]. The targeted temperature management (TTM) strategy includes a range of care aimed at actively managing the temperature according to the best goal. Therefore, hypothermia on admission and hyperthermia should be avoided, and the use of mild hypothermia is considered for patients with refractory intracranial hypertension. Measurement should be continuous and central (vesical or esophageal). The incorporation of brain temperature monitoring may provide additional information (1 or 2 °C higher than core temperature), allowing for reduced treatment intensity. However, it is not recommended to use brain temperature monitoring alone to guide treatment due to the lack of evidence. In all patients with TBI, fever above 38 °C should be treated. It is reasonable to actively target normothermia (36–37.5 °C) in patients who have required first- or second-tier measures for ICP control [76]. Hypothermia is effective in reducing ICP, but it is associated with adverse effects [77]. We propose mild hypothermia to a lower limit of 35 °C as a third-tier intervention. Induction should be rapid, avoiding daily temperature fluctuations greater than 1 °C. A summary of the main clinical scenarios is included in Table 3. Whilst there is no consensus, the first 5–7 days are considered the period of greatest vulnerability to secondary injury. Rewarming should be gradual (less than 0.25 °C per hour) to avoid rebound hyperthermia. In this context, careful monitoring of other variables that may influence ICP (PaCO_2_, sedation level, shivering) during rewarming is essential. The method used will depend on the intensity of therapy and local protocols, with invasive or surface systems being more effective [78]. Due to its modest effect in these cases, the use of NSAIDs and acetaminophen should be limited to stable patients with mild hyperthermia. Continuous infusion may be used to reduce the incidence of side effects. The use of a care bundle with measurement of hyperthermia burden is recommended to ensure high-quality TTM.

### 2.12. Devastating Brain Injury on Admission: What Does It Consist of and How to Deal with It?

Brain injuries may pose an immediate threat to life or unlikely functional recovery. These injuries are considered devastating brain injuries (DBIs) [79]. Massive hemispheric swelling, traumatic brainstem injury, or corpus callosum lesions are just a few examples of possible DBI [80]. In the initial phases of the approach, we will observe a low level of consciousness as well as pupillary changes (fixed mydriatic or anisochoric pupils) [81]. A DBI can be diagnosed during ICU care for a severely injured patient. In some cases, the diagnosis is made before ICU admission, using this period for stabilization and observation to develop an end-of-life care plan [82]. Accurate prognosis in the early stages of DBI cases can be difficult, even for experienced clinicians, and evidence-based prognostic strategies are limited [79,83]. Priority in the first hours is to maintain homeostasis, considering that the optimal period of observation is not well established. Some scientific societies suggest an observation period of 72 h, and others indicate that this period should be sufficient to allow a prognostic evaluation based on a combination of clinical judgment, changes in neurological function, degree of invasive support, and integration of the patient’s values and preferences [81]. Brain death may be the most common outcome, but in the case of persistent neurological damage, the multidisciplinary team that cares for the patient may consider adapting the therapeutic effort, given the futility of it. In both scenarios (brain death or withdrawal life-support therapy), we must approach the family to provide support and facilitate decision-making in the end-of-life care plan, including donation. In such circumstances, the organ procurement and transplantation unit will be contacted to evaluate donation opportunities, and it will be the transplant coordinator who offers the possibility of donation within the end-of-life care plan.

## Figures and Tables

**Figure 1 jcm-13-07325-f001:**
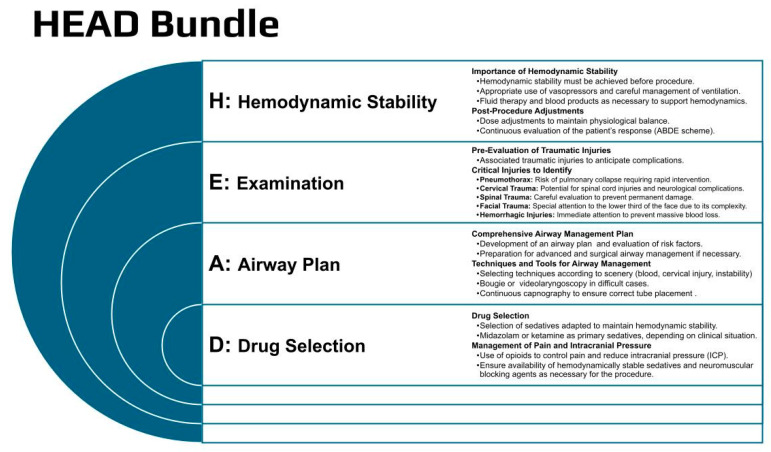
The HEAD bundle for intubation.

**Figure 2 jcm-13-07325-f002:**
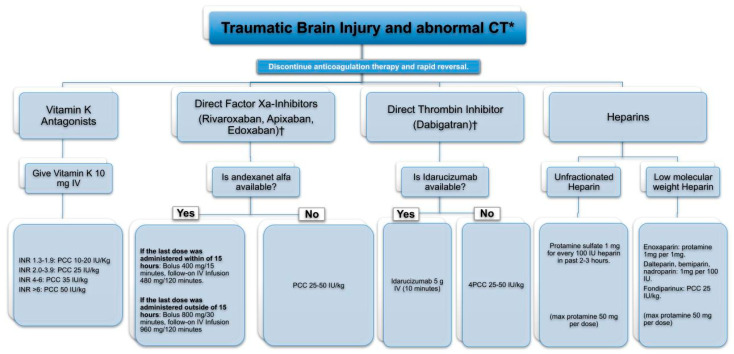
Approach to reversal of anticoagulation in traumatic brain hemorrhage. PCC: four-factor prothrombin complex concentrate; INR: international normalized ratio; CT: computed tomography; max: maximum; IU: international units; g: grams; mg: milligrams. * In patients with TBI and normal CT, it is advisable to discontinue anticoagulation, perform an observation period, and repeat CT before restarting anticoagulation. † Activated charcoal is used in spontaneous cerebral hemorrhage if the last intake is less than 8 h. In traumatic brain hemorrhage, it may interfere with the initial care of major trauma.

**Figure 3 jcm-13-07325-f003:**
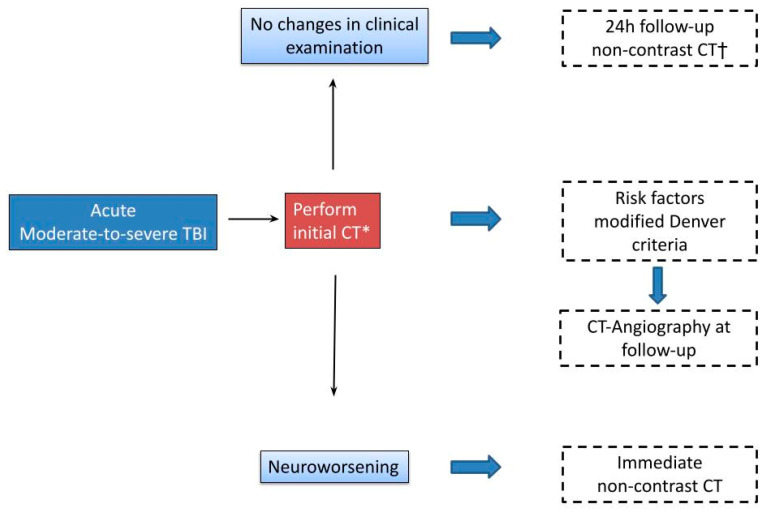
Neuroimaging workflow in moderate-to-severe TBI. * If DAI is suspected (unexplained neurological findings), schedule an MRI. † In patients with early (<2–3 h from injury) initial pathological CT scan, consider earlier follow-up.

**Table 1 jcm-13-07325-t001:** Risk factors for neuroworsening after TBI.

Neuroworsening Risk After TBI						
	GCS					
	14–15		9–13		3–8	
Marshall CT classification	Marshall I–II	Marshall > II	Marshall I–II	Marshall > II	Marshall I–II	Marshall > II
No risk factors						
Risk factors						
Risk factors						
Patient: age, comorbidities, anticoagulant use, alcohol, or drugs. Injury: extracranial injuries, pupillary abnormalities, hypotension, or hypoxia. Imaging: MLS > 5 mm, brainstem lesion, deep injuries, and ONSD > 5 mm. Complications: seizures, brain ischemia, or vascular injury.						
Adapted from the traffic light method proposed by Godoy et al. [5]. Color category description: Green: observation; Yellow: high risk; and Red: imminent risk. MLS: midline shift. ONSD: optic nerve sheath diameter.						

**Table 2 jcm-13-07325-t002:** Principal analgesics and sedatives for TBI.

Principal Analgesic and Sedatives for TBI					
	Dosis	Sedative Analgesic	ICP	MAP	Tachyphylaxis and Tolerance
**Midazolam**	**Bolus**: 0.02–0.08 mg/kg **Continuous infusion**: 0.04–0.30 mg/kg/h	**+++/−**	**↓**	**↓**	**↑↑↑**
**Propofol**	**Bolus**: 0.5–1 mg/kg **Continuous infusion**: 5–200 μg/kg/min	**+++/−**	**↓↓**	**↓↓**	**↑↑**
**Dexmedetomidine**	**Bolus**: Not recommended **Continuous infusion**: 0.4–1.2 μg/kg/h	**++/+**	**↔**	**↓**	**↔**
**Fentanyl**	**Bolus**: 25–125 μg **Continuous infusion**: 10–100 μg/h	**+/+++**	**↔**	**↓**	**↑↑**
**Remifentanil**	**Bolus**: Not recommended **Continuous infusion**: 0.05–0.25 μg/kg/min	**+/+++**	**↔**	**↓↓**	**↑↑**
**Morphine**	**Bolus**: Not recommended **Continuous infusion**: 0.02–0.07 mg/kg/h	**−/+++**	**↔**	**↓**	**↑↑↑**
**Tiopental**	**Bolus**: 2–3 mg/kg (20 sg) **Continuous infusion**: 3 mg/Kg/h	**+++/−**	**↓↓↓**	**↓↓↓**	**↔**
**Ketamine**	**Bolus**: Not recommended **Continuous infusion**: 1–5 mg/kg/h	**+++/++**	**↔**	**↑↑↑**	**↔**
**Isoflurane**	**Bolus or** **continuous administration**: Not recommended	**+++/−**	**↑↑↑**	**↓**	**↔**

↑↑ moderate increase; ↑↑↑ pronounced increase; ↔ no clear effect; ↓ modest decrease; ↓↓ moderate decrease; ↓↓↓ pronounced decrease; +++ pronounced effect; ++ moderate effect; + modest effect; − no effect. ICP: intracranial pressure; MAP: mean arterial pressure. Adapted from: Layon AJ [65].

**Table 3 jcm-13-07325-t003:** The continuation of TTM in TBI.

The Continuation of TTM in TBI			
Clinical Scenario	Admission Hypothermia	Fever and Hyperthermia	High ICP
**Recommended attitude**	Treat below 35 °C	Vigorous efforts to achieve normothermia (36–37.5) in the first 5–7 days	Mild hypothermia to a lower limit of 35 °C (third-tier intervention)
**How to achieve**	Passive rewarming	Active methods optional (surface or invasive)	Active methods desirable (surface or invasive)
**Comments**	Avoid rebound hyperthermia	In patients with Tier 1 or 2 for ICP	Review other causes of neuroworseing
**The not-to-do list for TTM** Prophylactic hypothermia. Deep hypothermia (below 35 °C). Not treating the shivering. Rapid rewarming and hyperthermia rebound. Lack of a temperature management protocol.			

## Data Availability

Not applicable.

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
