# Peer review of "Resuscitation and Initial Management After Moderate-to-Severe Traumatic Brain Injury: Questions for the On-Call Shift"

_jcm, 2024, doi:10.3390/jcm13237325_

Round 1
Reviewer 1 Report
Comments and Suggestions for Authors
Traumatic brain injury (TBI) constitutes one of the leading causes of disability and mortality worldwide and is important social problem.
Review article provides a comprehensive guide for the resuscitation and stabilization of TBI patients, combining clinical experience with current evidence-based guidelines. This practical review integrates foundational and recent advances in TBI management to aid in reducing secondary injuries and enhancing long-term recovery, presenting a multidisciplinary approach to support acute care decisions in TBI patients.
The content of the article aims to combine the best available evidence with bedside clinical experience in response to common challenges in the early hours following moderate to severe TBI.
The following questions for on-call shift are dealt with in the literature review:
1. Identification of severe or complicated TBI: scores and predictors of deterioration
The authors point to the limitation of the Glasgow coma scale (GCS) in making a forecast and write about the need for additions. Moderate TBI is allocated to a special risk group that requires special treatment (high risk of neuroworsening, requiring imaging studies and close neuromonitoring. Neurological examinations can be scheduled every 30 minutes for the first 2 hours). A table of risk factors is provided (Table 1), including coma scale GCS and an assessment of the Marshall CT classificftion scale. From our point of view, it is necessary to decipher the meaning of different colors and explain what a software assessment is Marshall CT.
2. Intubation in Brain Trauma: who, how, and when?
The authors emphasize the following important points: The intubation process can lead to dramatic increases in secondary damage due to hypoxia, hypotension, and elevated intracranial pressure (ICP); GCS of < 9 has been commonly used as a threshold a fixed limit is not recommended; Ensuring hemodynamic and ventilatory stability throughout the intubation process are critical; The use of sedatives that provide greater stability should be prioritized (midazolam or ketamine). Adding opioids can reduce ICP during airway manipulation.
3. What are the resuscitative treatment goals for this patient?
The authors emphasize the following important points: Blood pressure, cerebral perfusion pressure (CPP) and cerebral blood flow are interdependent variables that shape cerebral perfusion. Minimizing hypovolemia and hypotension are essential in preventing secondary brain damage. Given recent reports indicating higher mortality with balanced solutions in patients with acute brain injury, saline 0.9% should remain the resuscitation fluid of choice until new evidence emerges. PaCO2 is the main determinant of cerebral blood flow. Guidelines recommend PaCO2 between 35-45 mm Hg due to the risk of inducing ischemia.
4. How whould we manage patients with concomitant hemorrhagic shock and Traumatic Brain
Injury?
The authors emphasize the following important points: Hemorrhagic shock and TBI are the leading causes of morbidity and mortality in severe trauma. The coexistence of both in the same patient were observed in 3.8% of the patients and these patients had a mortality rate of 40%, with intracranial hypertension being the leading cause. A strategy of careful brain-focused resuscitation should be followed. This should include using higher blood pressure targets (SBP > 110 mmHg), avoiding hypotension, judicious use of sedation, and maintaining oxygenation, alongside aggressive correction of coagulopathy(hyperosmolar agents or mild hyperventilation might be used).
5. Coagulopathy in Traumatic Brain Injury: Diagnosis and Treatment
The authors emphasize the following important points: Coagulopathy in TBI correlates with injury severity as well as factors like cerebral hypoperfusion, hypoxia, and systemic inflammation, which exacerbate neuronal damage and increase the risk of secondary intracranial hemorrhage. We must adhere to general guidelines avoiding hypothermia and acidosis. In patients with TBI who develop coagulopathy, the use of fresh frozen plasma (FFP) or prothrombin complex concentrate (PCC) is recommended depending on the clinical situation. The administration of platelets in TBI aims to correct platelet dysfunction is 200 recommended for patients with significant intracranial hemorrhages.
6. How should we manage TBI patients who are on antiplatelet and anticoagulant therapy?
The authors emphasize the following important points: Specific phenotypes have been identified in which age and anticoagulant use correlate with poorer outcomes, even among patients exhibiting high scores on the GCS. Figure 2 shows the treatment regimen «Approach to reversal of anticoagulation in traumatic brain hemorrhage». The indication for platelet transfusion in patients with TBI is reserved for those with thrombocytopenia (less than 100 × 10^9/L), and for patients requiring urgent neurosurgical procedures.
7. Imaging: What tests and when are they needed?
The authors emphasize the following important points: Non-contrast CT is sensitive and specific for the presence of intracranial hemorrhage, extraaxial fluid collections, skull fractures, and radiopaque foreign bodies. Non-contrast CT also detects cerebral contusions, cerebral edema, swelling and signs of herniation. In suspected DAI (unexplained neurological findings despite “negative” non-contrast CT), the potential corpus callosum or brainstem involvement make magnetic resonance imaging (MRI) the elective imaging technique to diagnose and characterize the extent of DAI. Figure 3 demonstrate “ Neuroimaging workflow in moderate-to-severe TBI”.
8. Emergent neurosurgery: indications and controversies
The authors emphasize the following important points: Intracranial hematomas are present in 25% of patients with TBI who require hospital admission and up to 45% of patients with TBI who require ICU admission. Surgical evacuation is recommended if the hematoma volume is > 25-30 cc for epidural hematomas and brain contusions or thickness > 10 mm for acute subdural hematomas (SDH). Decompressive craniectomy can be helpful to decrease mortality. The placement of an external ventricular catheter is considered useful for reducing ICP but difficulties related to small ventricles.
9. Multimodal neuromonitoring: Who and how?
The authors emphasize the following important points: There is agreement on monitoring ICP in patients with a GCS < 9 after resuscitation and pathological CT. Therapeutic goals should be flexible and adapted to the patient’s physiology (e.g., ICP of 25 mmHg and normal PbtO2). It is essential to highlight the debate about the effect of neuromonitoring on patient outcomes. Discontinuation is recommended after at least 48 hours of meeting treatment targets and stable CT findings.
10. Analgesia and sedation: best decision in acute care of traumatic brain injury
The authors emphasize the following important points: Analgesia and sedation is an essential neurocritical care intervention, both at basic and advanced levels. The choice of the initial sedation and analgesia should be based on the pharmacokinetic and pharmacodynamic properties of the drugs used (see Table 2). Recent studies have shown that dexmedetomidine may have a neuroprotective effect. Adequate analgesia, using both opioid and non-opioid drugs, is crucial to reduce nociceptive stimuli that may increase ICP. Early discontinuation may trigger intracranial hypertension, leading to the harmful effects of secondary injury, while delayed withdrawal may increase the risk of complications such as infections, ICU-acquired weakness, delirium, and prolonged ICU stay.
11. Not too cold, not too hot: targeted temperature management
The authors emphasize the following important points: Hypo- and hyperthermia have been consistently associated with worse clinical outcomes in patients with TBI. The use of mild hypothermia is considered for patients with refractory intracranial hypertension. In all patients with TBI, fever above 38 ºC should be treated. It is reasonable to actively target normothermia (36-37.5ºC) in patients who have required first . or second-tier measures for ICP control. Table 3 “The continuous of TTM in TBI” provides recommendations for different levels of temperature monitoring.
12. Devastating brain injury on admission: ¿What does it consist of and how to deal with it?
The authors emphasize the following important points: Massive hemispheric swelling, traumatic brainstem injury or corpus callosum lesions are just a few examples of possible devastating brain injury (DBI). Priority in the first hours is to maintain homeostasis, considering that the optimal period of observation is not well established. Some scientific societies suggest an observation period of 72 hours and others indicate that this period should be sufficient to allow a prognostic evaluation based on a combination of clinical judgment, changes in neurological function, degree of invasive support, and integration of the patient's values and preferences. Brain death may be the most common outcome, but in the case of persistent neurological damage, the multidisciplinary team that cares for the patient may consider adapting the therapeutic effort. In both scenarios (brain death or withdrawal life-support therapy) we must approach the family to provide support and facilitate decision-making in the end-of-life care plan, including donation.
The authors of the article, most likely, based on their own experience and using data from the literature on the treatment of patients with TBI, described 12 important points, starting with assessing the severity of brain damage, treatment tactics for complications and ending with making decisions about treatment.
The article is of interest both for doctors working in intensive care units and for a wider range of researchers in the physiology and treatment of TBI. The article is illustrated with 3 tables and 3 figures. There is a local 1 reference to their own work in the list of 70 (2019 – 2022 years) references.
The only remark concerns Figure 1. It seems to me that it is necessary to decipher the values of the different colors in the diagram.
Author Response
We appreciate the reviewers comments. We have responded them and we believe that the quality of the manuscript has improved. Please, note that minor corrections have been made in figures as requested and that the length of the manuscript and the number of references has increased. We hope that this new version could be accepted in J Clin Med.
REVIEWER 1
Traumatic brain injury (TBI) constitutes one of the leading causes of disability and mortality worldwide and is important social problem.
Review article provides a comprehensive guide for the resuscitation and stabilization of TBI patients, combining clinical experience with current evidence-based guidelines. This practical review integrates foundational and recent advances in TBI management to aid in reducing secondary injuries and enhancing long-term recovery, presenting a multidisciplinary approach to support acute care decisions in TBI patients.
The content of the article aims to combine the best available evidence with bedside clinical experience in response to common challenges in the early hours following moderate to severe TBI.
The following questions for on-call shift are dealt with in the literature review:
- Identification of severe or complicated TBI: scores and predictors of deterioration
The authors point to the limitation of the Glasgow coma scale (GCS) in making a forecast and write about the need for additions. Moderate TBI is allocated to a special risk group that requires special treatment (high risk of neuroworsening, requiring imaging studies and close neuromonitoring. Neurological examinations can be scheduled every 30 minutes for the first 2 hours). A table of risk factors is provided (Table 1), including coma scale GCS and an assessment of the Marshall CT classificftion scale. From our point of view, it is necessary to decipher the meaning of different colors and explain what a software assessment is Marshall CT.
- Intubation in Brain Trauma: who, how, and when?
The authors emphasize the following important points: The intubation process can lead to dramatic increases in secondary damage due to hypoxia, hypotension, and elevated intracranial pressure (ICP); GCS of < 9 has been commonly used as a threshold a fixed limit is not recommended; Ensuring hemodynamic and ventilatory stability throughout the intubation process are critical; The use of sedatives that provide greater stability should be prioritized (midazolam or ketamine). Adding opioids can reduce ICP during airway manipulation.
- What are the resuscitative treatment goals for this patient?
The authors emphasize the following important points: Blood pressure, cerebral perfusion pressure (CPP) and cerebral blood flow are interdependent variables that shape cerebral perfusion. Minimizing hypovolemia and hypotension are essential in preventing secondary brain damage. Given recent reports indicating higher mortality with balanced solutions in patients with acute brain injury, saline 0.9% should remain the resuscitation fluid of choice until new evidence emerges. PaCO2 is the main determinant of cerebral blood flow. Guidelines recommend PaCO2 between 35-45 mm Hg due to the risk of inducing ischemia.
- How whould we manage patients with concomitant hemorrhagic shock and Traumatic Brain Injury?
The authors emphasize the following important points: Hemorrhagic shock and TBI are the leading causes of morbidity and mortality in severe trauma. The coexistence of both in the same patient were observed in 3.8% of the patients and these patients had a mortality rate of 40%, with intracranial hypertension being the leading cause. A strategy of careful brain-focused resuscitation should be followed. This should include using higher blood pressure targets (SBP > 110 mmHg), avoiding hypotension, judicious use of sedation, and maintaining oxygenation, alongside aggressive correction of coagulopathy (hyperosmolar agents or mild hyperventilation might be used).
- Coagulopathy in Traumatic Brain Injury: Diagnosis and Treatment
The authors emphasize the following important points: Coagulopathy in TBI correlates with injury severity as well as factors like cerebral hypoperfusion, hypoxia, and systemic inflammation, which exacerbate neuronal damage and increase the risk of secondary intracranial hemorrhage. We must adhere to general guidelines avoiding hypothermia and acidosis. In patients with TBI who develop coagulopathy, the use of fresh frozen plasma (FFP) or prothrombin complex concentrate (PCC) is recommended depending on the clinical situation. The administration of platelets in TBI aims to correct platelet dysfunction is 200 recommended for patients with significant intracranial hemorrhages.
- How should we manage TBI patients who are on antiplatelet and anticoagulant therapy?
The authors emphasize the following important points: Specific phenotypes have been identified in which age and anticoagulant use correlate with poorer outcomes, even among patients exhibiting high scores on the GCS. Figure 2 shows the treatment regimen «Approach to reversal of anticoagulation in traumatic brain hemorrhage». The indication for platelet transfusion in patients with TBI is reserved for those with thrombocytopenia (less than 100 × 10^9/L), and for patients requiring urgent neurosurgical procedures.
- Imaging: What tests and when are they needed?
The authors emphasize the following important points: Non-contrast CT is sensitive and specific for the presence of intracranial hemorrhage, extraaxial fluid collections, skull fractures, and radiopaque foreign bodies. Non-contrast CT also detects cerebral contusions, cerebral edema, swelling and signs of herniation. In suspected DAI (unexplained neurological findings despite “negative” non-contrast CT), the potential corpus callosum or brainstem involvement make magnetic resonance imaging (MRI) the elective imaging technique to diagnose and characterize the extent of DAI. Figure 3 demonstrate “ Neuroimaging workflow in moderate-to-severe TBI”.
- Emergent neurosurgery: indications and controversies
The authors emphasize the following important points: Intracranial hematomas are present in 25% of patients with TBI who require hospital admission and up to 45% of patients with TBI who require ICU admission. Surgical evacuation is recommended if the hematoma volume is > 25-30 cc for epidural hematomas and brain contusions or thickness > 10 mm for acute subdural hematomas (SDH). Decompressive craniectomy can be helpful to decrease mortality. The placement of an external ventricular catheter is considered useful for reducing ICP but difficulties related to small ventricles.
- Multimodal neuromonitoring: Who and how?
The authors emphasize the following important points: There is agreement on monitoring ICP in patients with a GCS < 9 after resuscitation and pathological CT. Therapeutic goals should be flexible and adapted to the patient’s physiology (e.g., ICP of 25 mmHg and normal PbtO2). It is essential to highlight the debate about the effect of neuromonitoring on patient outcomes. Discontinuation is recommended after at least 48 hours of meeting treatment targets and stable CT findings.
- Analgesia and sedation: best decision in acute care of traumatic brain injury
The authors emphasize the following important points: Analgesia and sedation is an essential neurocritical care intervention, both at basic and advanced levels. The choice of the initial sedation and analgesia should be based on the pharmacokinetic and pharmacodynamic properties of the drugs used (see Table 2). Recent studies have shown that dexmedetomidine may have a neuroprotective effect. Adequate analgesia, using both opioid and non-opioid drugs, is crucial to reduce nociceptive stimuli that may increase ICP. Early discontinuation may trigger intracranial hypertension, leading to the harmful effects of secondary injury, while delayed withdrawal may increase the risk of complications such as infections, ICU-acquired weakness, delirium, and prolonged ICU stay.
- Not too cold, not too hot: targeted temperature management
The authors emphasize the following important points: Hypo- and hyperthermia have been consistently associated with worse clinical outcomes in patients with TBI. The use of mild hypothermia is considered for patients with refractory intracranial hypertension. In all patients with TBI, fever above 38 ºC should be treated. It is reasonable to actively target normothermia (36-37.5ºC) in patients who have required first or second-tier measures for ICP control. Table 3 “The continuous of TTM in TBI” provides recommendations for different levels of temperature monitoring.
- Devastating brain injury on admission: ¿What does it consist of and how to deal with it?
The authors emphasize the following important points: Massive hemispheric swelling, traumatic brainstem injury or corpus callosum lesions are just a few examples of possible devastating brain injury (DBI). Priority in the first hours is to maintain homeostasis, considering that the optimal period of observation is not well established. Some scientific societies suggest an observation period of 72 hours and others indicate that this period should be sufficient to allow a prognostic evaluation based on a combination of clinical judgment, changes in neurological function, degree of invasive support, and integration of the patient's values and preferences. Brain death may be the most common outcome, but in the case of persistent neurological damage, the multidisciplinary team that cares for the patient may consider adapting the therapeutic effort. In both scenarios (brain death or withdrawal life-support therapy) we must approach the family to provide support and facilitate decision-making in the end-of-life care plan, including donation.
The authors of the article, most likely, based on their own experience and using data from the literature on the treatment of patients with TBI, described 12 important points, starting with assessing the severity of brain damage, treatment tactics for complications and ending with making decisions about treatment.
The article is of interest both for doctors working in intensive care units and for a wider range of researchers in the physiology and treatment of TBI. The article is illustrated with 3 tables and 3 figures. There is a local 1 reference to their own work in the list of 70 (2019 – 2022 years) references.
The only remark concerns Figure 1. It seems to me that it is necessary to decipher the values of the different colors in the diagram.
We have added the following color categorization similarly to the article by Godoy et al: Color categories description: Green: Observation; Yellow: High risk; Red: Imminent risk.
Reviewer 2 Report
Comments and Suggestions for Authors
Thank you for letting me comment on this interesting review regarding the current issues of traumatic brain injury in adults. In general, this manuscript describes the basic considerations that a clinician should have in mind regarding TBI, including some interesting algorithms that might prove useful in clinical practice. However, some major and some minor modifications are necessary.
Major modifications:
1) Please recheck the whole manuscript with regard to bibliographic references, because there are many statements that are not based on literature, and it is not clear if they should be considered as the authors' expert opinion.
2) Figure 2: Please correct the Sentences of the HEAD bundle, because besides H and A, the other letters do not make sense for the support of the acronym (e.g. the sentence for E could begin with "examination" and for D the word drugs could be used alternatively)
Minor modifications:
1) Line 36: Please change into "neurosurgical interventions"
2) Line 79: Please elaborate a little bit more on the radiological scores and biomolecules currently available for TBI classification.
3) Line 82: Please provide the current literature for the statement in the sentence "neurological examinations....intervals"
4) Table 1: Legend--> Please provide in parenthesis the citation of Godoy
5) Line 98: Please clarify that you refer to bulbar (brainstem) reflexes, not neurological reflexes in general
6) Line 104: Please explain in parenthesis what C-ABC is, so that someone who is not an expert on life support can follow
7) Lines 105-106: Please provide the bibliographic citation for this sentence.
8) Lines 112-115: Please provide the bibliographic citation for this sentence.
9) Lines 152: Please provide another citation for supporting this sentence, other than the RETRAUCI registry.
10) Line 168: Please correct the typo “individualized” into “individualize”
11) Line 211: Please comment a little bit on why TXA is not recommended in severe TBI, so that the reader can follow.
12) Line 218: Please correct the typo “in which age”
13) Figure 2: Please correct the typos “Rivaroxban”and “Enoxaparina”
14) Figure 3: What is “Stat non-contrast CT”??
15) Line 292: Please correct the typo “intracranial hematomas”
16) Line 322: Correct the typo “¿which patients”
17) Line 338: Correct the typo “(ACTRN12619001328167P, 338 NCT03754114);”
18) Line 341: What is SIBICC stand for?
19) Line 375: Correct the typo in parenthesis
20) Line 423: Please correct “¿”
I am looking forward to your modifications!

Author Response
REVIEWER 2
Thank you for letting me comment on this interesting review regarding the current issues of traumatic brain injury in adults. In general, this manuscript describes the basic considerations that a clinician should have in mind regarding TBI, including some interesting algorithms that might prove useful in clinical practice. However, some major and some minor modifications are necessary.
Major modifications:
1) Please recheck the whole manuscript with regard to bibliographic references, because there are many statements that are not based on literature, and it is not clear if they should be considered as the authors' expert opinion. We have added new references to support several sentences as requested.
2) Figure 2: Please correct the Sentences of the HEAD bundle, because besides H and A, the other letters do not make sense for the support of the acronym (e.g. the sentence for E could begin with "examination" and for D the word drugs could be used alternatively). The reviewer is right. We have modified the figure in accordance with the suggestion of the reviewer.
Minor modifications:
1) Line 36: Please change into "neurosurgical interventions". Corrected.
2) Line 79: Please elaborate a little bit more on the radiological scores and biomolecules currently available for TBI classification. The sentence has been expanded and now it reads as follows: In this context, a stratification system that incorporates clinical evaluation, imaging (Marshall, Rotterdam and Helsinki classifications), and ideally, biomolecular markers (mainly glial fibrillary acidic protein (GFAP) and ubiquitin C-terminal hydrolase L1 (UCH-L1)) could help to further individualize treatment.
3) Line 82: Please provide the current literature for the statement in the sentence "neurological examinations....intervals". The reviewer is right. This recommendation came from articles including patients with ischemic stroke and thrombolysis, so we have removed this sentence following hir/her valuable comment.
4) Table 1: Legend--> Please provide in parenthesis the citation of Godoy. The complete citation has been added in the legend of table 1.
5) Line 98: Please clarify that you refer to bulbar (brainstem) reflexes, not neurological reflexes in general. The reviewer is right. We have added “bulbar” reflexes.
6) Line 104: Please explain in parenthesis what C-ABC is, so that someone who is not an expert on life support can follow. This sentence has been rephrased to “Depending on the patient's condition, C-ABC resequencing (Catastrophic compressible hemorrhage - Airway; Breating; Circulation), may be necessary to prevent hemodynamic collapse”.
7) Lines 105-106: Please provide the bibliographic citation for this sentence. We have completed the sentence and added this recent relevant reference: This involves the early use of vasopressors, availability of blood products, and rapid screening for associated injuries (ie. pneumothorax) prioritizing circulation over intubation.
8) Lines 112-115: Please provide the bibliographic citation for this sentence. We have added the reference of the guidelines for the management of trachel intubation in critically ill adults.
9) Lines 152: Please provide another citation for supporting this sentence, other than the RETRAUCI registry. We have added the following text from a recent study and included the new reference. In a recent study including 564 patients with TBI and hemorrhagic shock evaluating the effect of whole blood transfusion, overall 30-day mortality was 42% (237/564). Death was attributed to TBI in 100/237 cases (42.2%)
10) Line 168: Please correct the typo “individualized” into “individualize”. Corrected.
11) Line 211: Please comment a little bit on why TXA is not recommended in severe TBI, so that the reader can follow. We have added the main results of the landmark CRASH-3 trial to support this sentence. Now it reads as follows: According to the landmark CRASH-3 randomized controlled trial, the risk of head injury-related death was reduced with tranexamic acid in patients with mild-to-moderate head injury (RR 0·78 [95% CI 0·64-0·95]) but not in patients with severe head injury (0·99 [95% CI 0·91-1·07]; p value for heterogeneity 0·030), so its use is not routinely recommended in cases with severe TBI.
12) Line 218: Please correct the typo “in which age”. The sentence has been split into 2 shorter sentences. Now it reads as follows: Specific phenotypes have been identified. Age and anticoagulant use correlate with poorer outcomes.
13) Figure 2: Please correct the typos “Rivaroxban”and “Enoxaparina”. Corrected.
14) Figure 3: What is “Stat non-contrast CT”?? This special designation is meant to be used in case of an emergency where the results of a test or procedure may mean the difference between life and death. Since this might be less-known, we have changed it to” Immediate”.
15) Line 292: Please correct the typo “intracranial hematomas”. Corrected
16) Line 322: Correct the typo “¿which patients”. Corrected
17) Line 338: Correct the typo “(ACTRN12619001328167P, 338 NCT03754114);”. We have reviewed the trials code appearing in our version and correspond to the BONANZA and BOOST-3 trials. Maybe in the reviewer version is there a mistake. It looks that the number of the line 338 appears in between the codes.
18) Line 341: What is SIBICC stand for? We have specified the meaning of SIBBIC as follows: The Seattle International Severe Traumatic Brain Injury Consensus Conference (SIBICC) guidelines.
19) Line 375: Correct the typo in parenthesis. We have reviewed the code and corresponds to the BIKe (Brain Injury and Ketamine) study as we wanted to do.
20) Line 423: Please correct “¿”. Corrected
I am looking forward to your modifications! We thank the reviewer for his/her constuctive criticisms
Reviewer 3 Report
Comments and Suggestions for Authors
The authors made a review of the very wide subject : management of the TBI. They have mentioned many aspects of the challenging treatment but they remain superficial in most of it. This is not surprising as others wrote whole books on this subject, a 10 pages review is certainly not enough. The manuscript is good readable and understandable but lacks in novelty. It is rather a guide for young clinicians with important first-line information.
I would like to address some points of potential improvement:
You talk about the intubation but 1) you do not refer to the position of the patient when intubated -> normally the head is lowered in order to facilitate the intubation but in a patient with TBI and brain oedema this could be fatal, 2) sedatives should be avoided when possible as they alter the mental status and the GCS leading probably the surgical colleagues to wrong decisions. 3) you should refer to alternative methods when a traditional intubation is due to the severe injuries not possible. 4) when should an intubation take place? The timing is very important, sometimes the decision between experience of the intubating doctor and the GCS score is critical. Wait for an experienced doctor to do the difficult intubation and risk the health of the patient or take the risk of doing a wrong intubation (oesophageal, one sided etc).
The administration of FFP has a short effect (about 2 hours) and should be favored when surgery is planned.
Furthermore, you should discuss the thrombotic risk in those patients, they are immobilized and receive thrombocytes and TXA.
A figure with the workflow in a TBI patient admission would be nice.
You talk about the progression of a bleeding or other findings the first 6-9 hours but you did not mention your opinion about follow-up examination after 6-8 hours.
Please talk about the cases where the patient is too hemodynamic-unstable to do a CT scan or be transported.
Refer to extra CT-venography in patients with fractures above the venous sinuses.
Epidural hematomas tend to expand rapidly due to the meningeal artery and for many are a direct surgical indication. The size of the acute subdural hematomas is relative. 1cm wide hematoma in a patient over 70 has very little impact due to the brain atrophy and is not an indication. So please revise this part of the text, the indications you report are not right. We base our indication on the neurological condition of the patient and on the impact of the findings on the brain.
Hematomas with lower volume do not cause midline shift and when they do, it is the oedema causing it, not the 1cm blood line.
Your really have to work with an experienced neurosurgeon on 2.8. Many things you say are not right or misleading. Decompressive craniectomy is the only solution in patients with brain oedema. It may be less effective in patients with bleeding but without oedema but it is not what you write. An EVD (external ventricle drainage) is not indicated in very small ventricles as it may cause cerebellar herniation if the pathology is there. CSF infection is more a possible complication than the leak.
You should talk about delirium management in those patients.
GCS 9-13 is not an indication for invasive ICP monitoring. The patients are in most cases awake and cooperative.
You should talk about autoregulation in extend.
The prophylactic heparin should be discussed more, what and when should be administered with references, there are more than enough out there, especially reviews about early anticoagulation in TBI.
What about seizure diagnosis and management?
You write: In all patients with TBI, fever above 38 ºC should be treated -> what about central fever? Do you try to treat and how?
Author Response
REVIEWER 3
The authors made a review of the very wide subject: management of the TBI. They have mentioned many aspects of the challenging treatment but they remain superficial in most of it. This is not surprising as others wrote whole books on this subject, a 10 pages review is certainly not enough. The manuscript is good readable and understandable but lacks in novelty. It is rather a guide for young clinicians with important first-line information.
We agree with the reviewer. It is not possible to encompass all aspects of TBI management in a single practical review like ours. Each of the comments made by the reviewer might likely require an specific chapter. As pointed, we tried to provide a practical guide for less experienced physicians in a general journal like this.
I would like to address some points of potential improvement:
You talk about the intubation but 1) you do not refer to the position of the patient when intubated -> normally the head is lowered in order to facilitate the intubation but in a patient with TBI and brain oedema this could be fatal.
You are correct about the concern regarding the position and its impact on ICP. We mentioned this risk and recommended a ramped position to minimize this effect and reduce the risk of bronchial aspiration: “A ramped position may reduce the rise in ICP and the risk of bronchoaspiration”.
We added two references to support this statement:
McCormack E, Aysenne A, Cardona JJ, et al. Effects of intubation technique on intracranial pressure: a cadaveric study. Neurosurg Rev. 2023;46(1):88. doi:10.1007/s10143-023-01996-4
Rolighed Larsen JK, Haure P, Cold GE. Reverse Trendelenburg position reduces intracranial pressure during craniotomy. J Neurosurg Anesthesiol. 2002;14(1):16-21. doi:10.1097/00008506-200201000-00004
2) sedatives should be avoided when possible as they alter the mental status and the GCS leading probably the surgical colleagues to wrong decisions.
We agree on the impact of sedation during reevaluation. We recommend intubation and sedation only when the potential benefits outweigh the risks, such as improved oxygenation, prevention of bronchial aspiration, or management of an altered respiratory pattern. In figure 1 we advocate for a careful use of these agents.
3) you should refer to alternative methods when a traditional intubation is due to the severe injuries not possible. And 4) when should an intubation take place? The timing is very important, sometimes the decision between experience of the intubating doctor and the GCS score is critical. Wait for an experienced doctor to do the difficult intubation and risk the health of the patient or take the risk of doing a wrong intubation (oesophageal, one sided etc).
We agree with your concern. Experience (and adequate training) is one of the most critical factors in the successful management of the airway. To emphasize this, the paragraph has been modified and reads as follows:
“Professionals who perform intubation should be equipped to thoroughly assess the patient’s injuries while being proficient in advanced airway management following standardized practices. Experience and training are key to safely managing the emergency airway, since alternative methods, such as a laryngeal mask or surgical airway, should be employed to ensure oxygenation if necessary”.
The administration of FFP has a short effect (about 2 hours) and should be favored when surgery is planned. Furthermore, you should discuss the thrombotic risk in those patients, they are immobilized and receive thrombocytes and TXA.
We have added that “The effect of FFP may be transient, especially if the underlying cause of coagulopathy persists or if there is ongoing loss of coagulation factors due to active bleeding. Therefore, repeated doses of FFP are often required to maintain adequate hemostasis” and latter we have added this paragraph at the end of this section in response to the query of DVT prophylaxis:”In this complex context, there are also controversies and questions remaining regarding the timing, dosing and agent to use for venous thromboprophylaxis in TBI, a relevant issue in immobilized patients that may have received TXA”
A figure with the workflow in a TBI patient admission would be nice. You talk about the progression of a bleeding or other findings the first 6-9 hours but you did not mention your opinion about follow-up examination after 6-8 hours.
We have added the following comment (authors opinion): “During the initial resuscitation phase, we recommend that coagulation parameters should be reviewed every 4–6 hours. Subsequently, if the patient has stabilized, monitoring should be performed every 12–24 hours”.
Please talk about the cases where the patient is too hemodynamic-unstable to do a CT scan or be transported.
We have added the following paragraph with reference: “In cases of TBI with concurrent severe hemodynamic instability in the context of life-threatening hemorrhage, we must priotitize the precise and early control of hemorrhage with associated restoration of circulating blood volume. Uncontrolled cases may require simultaneous imaging and multisystem surgery in hybrid rooms, being the main objective the control of bleeding and the avoidance/minimization of secondary brain insults”.
Refer to extra CT-venography in patients with fractures above the venous sinuses. We have added the following paragraph: Additionally, traumatic venous injury can occur, although it is commonly overlooked. From the imaging point of view, the most important risk factor for traumatic venous injury is a skull fracture (or less commonly a penetrating foreign body) that involves a dural venous sinus or jugular bulb/foramen [38]. In these cases, the performance of a venous-CT is recommended [38].
Epidural hematomas tend to expand rapidly due to the meningeal artery and for many are a direct surgical indication. The size of the acute subdural hematomas is relative. 1cm wide hematoma in a patient over 70 has very little impact due to the brain atrophy and is not an indication. So please revise this part of the text, the indications you report are not right. We base our indication on the neurological condition of the patient and on the impact of the findings on the brain.
Hematomas with lower volume do not cause midline shift and when they do, it is the oedema causing it, not the 1cm blood line.
Your really have to work with an experienced neurosurgeon on 2.8. Many things you say are not right or misleading. Decompressive craniectomy is the only solution in patients with brain oedema. It may be less effective in patients with bleeding but without oedema but it is not what you write. An EVD (external ventricle drainage) is not indicated in very small ventricles as it may cause cerebellar herniation if the pathology is there. CSF infection is more a possible complication than the leak.
The complete section 2.8 has been written and reviewed after criticisms by experienced neurosurgeons (both in the clinical and academic settings) that were listed as co-authors. Please, find below the response to the queries listed above and the updated version of the section
In 2006, guidelines for the surgical management of traumatic brain injury (TBI) were published. These recommendations were based on case series that aimed to identify patients with potentially surgical lesions who might experience neurological deterioration if surgery was not promptly performed.
Below is a summary of key points from these guidelines:
“An epidural hematoma (EDH) greater than 30 cm3 should be surgically evacuated regardless of the patient's Glasgow Coma Scale (GCS) score. An EDH less than 30 cm3 and with less than a 15-mm thickness and with less than a 5-mm midline shift (MLS) in patients with a GCS score greater than 8 without focal deficit can be managed nonoperatively with serial computed tomographic (CT) scanning and close neurological observation in a neurosurgical center.” (Bullock MR, Chesnut R, Ghajar J, Gordon D, Hartl R, Newell DW, Servadei F, Walters BC, Wilberger JE; Surgical Management of Traumatic Brain Injury Author Group. Surgical management of acute epidural hematomas. Neurosurgery. 2006 Mar;58(3 Suppl):S7-15; discussion Si-iv. PMID: 16710967.)
“An acute subdural hematoma (SDH) with a thickness greater than 10 mm or a midline shift greater than 5 mm on computed tomographic (CT) scan should be surgically evacuated, regardless of the patient's Glasgow Coma Scale (GCS) score. All patients with acute SDH in coma (GCS score less than 9) should undergo intracranial pressure (ICP) monitoring. A comatose patient (GCS score less than 9) with an SDH less than 10-mm thick and a midline shift less than 5 mm should undergo surgical evacuation of the lesion if the GCS score decreased between the time of injury and hospital admission by 2 or more points on the GCS and/or the patient presents with asymmetric or fixed and dilated pupils and/or the ICP exceeds 20 mm Hg.” (Bullock MR, Chesnut R, Ghajar J, Gordon D, Hartl R, Newell DW, Servadei F, Walters BC, Wilberger JE; Surgical Management of Traumatic Brain Injury Author Group. Surgical management of acute subdural hematomas. Neurosurgery. 2006 Mar;58(3 Suppl):S16-24; discussion Si-iv. PMID: 16710968.)
“Patients with parenchymal mass lesions and signs of progressive neurological deterioration referable to the lesion, medically refractory intracranial hypertension, or signs of mass effect on computed tomographic (CT) scan should be treated operatively. Patients with Glasgow Coma Scale (GCS) scores of 6 to 8 with frontal or temporal contusions greater than 20 cm3 in volume with midline shift of at least 5 mm and/or cisternal compression on CT scan, and patients with any lesion greater than 50 cm3 in volume should be treated operatively. Patients with parenchymal mass lesions who do not show evidence for neurological compromise, have controlled intracranial pressure (ICP), and no significant signs of mass effect on CT scan may be managed nonoperatively with intensive monitoring and serial imaging.” (Bullock MR, Chesnut R, Ghajar J, Gordon D, Hartl R, Newell DW, Servadei F, Walters BC, Wilberger J; Surgical Management of Traumatic Brain Injury Author Group. Surgical management of traumatic parenchymal lesions. Neurosurgery. 2006 Mar;58(3 Suppl):S25-46; discussion Si-iv. doi: 10.1227/01.NEU.0000210365.36914.E3. PMID: 16540746.)
Although the 2006 guidelines have not been updated and are based on low levels of evidence, the Marshall CT classification system and subsequent studies have reinforced their utility. A hematoma volume of >25–30 mL remains a key threshold for surgical evacuation, significantly improving outcomes. Surgical evacuation is recommended for epidural hematomas (EDH) or brain contusions exceeding this volume, as the typical pressure-volume index of 25 mL indicates that even small additional volumes can significantly increase intracranial pressure (ICP). However, not all EDHs require evacuation. The risk of expansion depends on factors such as the origin of the bleeding (e.g., venous vs. arterial) and the timing of the initial CT scan relative to the injury.
For acute subdural hematomas (SDH), the crescent shape makes hematoma thickness a more reliable parameter than volume. In patients with brain atrophy, conservative management is possible for SDHs >10 mm if symptoms are mild, allowing the hematoma to chronify and be evacuated later via less invasive procedures. Conversely, younger patients with high-energy injuries may exhibit significant MLS disproportionate to hematoma volume. In these cases, surgical evacuation can reduce ICP and mitigate ischemic injury caused by focal mass effect.
Decompressive craniectomy (DC) remains a last-tier intervention for refractory ICP, per TBI expert consensus. However, modern ICU management has shown that early DC, before exhausting other ICP management options, does not improve mortality or functional outcomes (as per the DECRA study). Thus, DC should be considered only after less invasive measures have failed, factoring in the patient’s CT findings, brain oxygenation levels, and clinical condition.
The placement of an external ventricular drain (EVD) is a first-tier ICP management strategy (SIBICC consensus) but poses challenges in patients with small ventricles, coagulopathy, or concurrent cerebrospinal fluid (CSF) leaks secondary to skull base fractures as we mentioned in the manuscript. The risk of CNS infection is increased if a EVD is placed for more than 10 days and in patients with CSF leak (as CSF drainage through the EVD can promote the upward migration of pathogens from the ear and nose). However, the relationship between skull base fracture and EVD infection is unfrequently confirmed during clinical practice. Then, we omitted this information in the revised version of the manuscript.
In summary, we have revised the section to address some of your concerns, and we hope you find it more appropriate:
Intracranial hematomas are present in 25% of patients with TBI who require hospital admission and up to 45% of patients with TBI who require ICU admission [41]. Guidelines of the surgical management of TBI patients have not been updated since their publication and the recommendations were based on case series [42]. However, the natural history of most intracranial hematomas or high ICP states preclude the design of randomized controlled trials in many circumstances. In general, surgical evacuation is recommended if the hematoma volume is > 25-30 cc for epidural hematomas (EDH) and brain contusions (ICH) or thickness > 10 mm for acute subdural hematomas (SDH). Although these recommendations are made regardless of the patient’s level of consciousness, as clinical deterioration is expected without intervention, patients with brain atrophy may tolerate such volumes. Therefore, additional features, such as signs of mass effect and the patient’s neurological condition, must be considered. Lesions without significant mass effect and in patients with stable clinical conditions can often be managed conservatively. On the other hand, for hematomas with smaller volumes, surgical evacuation is still recommended if there is a midline shift >5 mm or effacement of the basal cisterns and/or in case of neurological worsening not explained by extracranial reasons. This scenario is commonly associated with the location of EDH or ICH (e.g., temporal region) or with SDH, which can contribute to ischemic changes in the affected hemisphere due to local mass effects. Surgery for hematomas located in the posterior fossa is indicated if there is distortion, or obliteration of the fourth ventricle, compression of the basal cisterns, or the presence of obstructive hydrocephalus. Patients with TBI may develop brain swelling and refractory high ICP although hematoma is evacuated. Secondary decompressive craniectomy (DC) can be helpful to decrease mortality when used after exhausted other treatment options. However, the benefit in mortality can be translated into additional survivors with disability [43]. The pre-emptive decision to not replace bone flap after SDH evacuation (primary DC) seemed not to be associated with reduction of mortality or better functional outcome [44]. The placement of an external ventricular drainage (EVD) is useful for reducing ICP. However, the risk of EVD-related infection, difficulties related to small ventricles and coagulopathy must be addressed before indication [45]. Surgical decision-making is a complex process in which additional characteristics should be accounted including patient’s age, comorbidities, antithrombotic treatment, frailty, pupillary asymmetry, timing from clinical deterioration and presence of severe hypoxia or shock which are important factors influencing patient’s outcome. Additionally, reviewing the patient's living will, either through official documentation or consultation with relatives, is essential, especially before recommending a surgical treatment that could result in survival with significant disability.
You should talk about delirium management in those patients. GCS 9-13 is not an indication for invasive ICP monitoring. The patients are in most cases awake and cooperative.
As you pointed out, patients with a GCS score of 9-13 do not have a clear indication for invasive monitoring. Therefore, we have rewritten the paragraph to tone-down such recommendation. Now it reads as follows:
“Some of the most difficult patients are those with GCS 9-13 and injuries that do not meet criteria for surgical evacuation. These patients are usually difficult to assess clinically (neurological fluctuation, restlessness) and do not have a clear indication for invasive monitoring. However, they may benefit from such invasive monitoring if risk factors for clinical deterioration or systemic conditions are present, in example, in those patients who do not clearly maintain their airway patency requiring sedation and mechanical ventilation”.
You should talk about autoregulation in extend. We have added the following paragraph in the resuscitate treatment goals section: “The Phase II COGiTATE trial demonstrated the feasibility and safety of individualized CPP management guided by cerebral autoregulation. The authors utilized the PRx, derived from the ICP and arterial pressure monitoring, to adjust CPP based on the concept of optimal CPP (defined as the CPP value corresponding to the lowest PRx value). While robust evidence is still lacking, this approach appears promising in tailoring optimal CPP ranges and potentially improving outcomes”.
The prophylactic heparin should be discussed more, what and when should be administered with references, there are more than enough out there, especially reviews about early anticoagulation in TBI.
We have added the following paragraph in response to the comment of the reviewer: “In this complex context, there are also controversies and questions remaining regarding the timing, dosing and agent to use for venous thromboprophylaxis in TBI, a relevant issue in immobilized patients that may have received TXA. The 2016 version of the Brain Trauma Foundation guidelines state that there is insufficient evidence to support a Level I or II recommendation for treatment of deep venous thrombosis (DVT) in severe TBI patients. Low molecular weight heparin (LMWH) or low-dose unfractioned heparin may be used in combination with mechanical prophylaxis. However, there is an increased risk for expansion of intracranial hemorrhage. In addition to compression stockings, pharmacologic prophylaxis may be considered if the brain injury is stable and the benefit is considered to outweigh the risk of increased intracranial hemorrhage. There is insufficient evidence to support recommendations regarding the preferred agent, dose, or timing of pharmacologic prophylaxis for DVT”
What about seizure diagnosis and management?
We have added the following paragraph: “Seizure prophylaxis constitutes another matter of debate in moderate to severe TBI. Recent evidence and recommendations from the Neurocritical Care Society following a population, intervention, comparator, and outcome (PICO) approach suggest that, based on GRADE criteria, prophylactic antiseizure medications (ASM) or no ASM may be used in patients hospitalized with moderate to severe TBI (weak recommendation, low quality of evidence). When ASM are used, they suggest using levetiracetam over phenytoin/fosphenytoin (weak recommendation, very low quality of evidence) for a short duration (≤ 7 days, weak recommendation, low quality of evidence) These recommendations are different to that from the 2016 Brain Trauma Foundation guidelines but are aligned with current clinical practice in the U.S.
You write: In all patients with TBI, fever above 38 ºC should be treated -> what about central fever? Do you try to treat and how?
To the best of the authors' knowledge, there are no differentiated thresholds for temperature management based on its origin (fever vs. hyperthermia) in the TBI population. Central fever shows little to no response to pharmacological treatment but similar effects of ICP, we only recommend using NSAIDs or acetaminophen in stable patients with mild hyperthermia. We have edited the sentence to: “Due to its modest effect in these cases, the use of NSAIDs and acetaminophen should be restricted to stable patients with mild hyperthermia.”
We thank the reviewer for his/her constructive criticisms

Round 2
Reviewer 3 Report
Comments and Suggestions for Authors
The changes substantially improved the manuscript.